# DexVIP: Learning Dexterous Grasping with Human Hand Pose Priors from Video

**Priyanka Mandikal**[1,2] **and Kristen Grauman**[1,2]
[1]Department of Computer Science, The University of Texas at Austin
[2]Facebook AI Research
{mandikal,grauman}@cs.utexas.edu

**Abstract:** Dexterous multi-fingered robotic hands have a formidable action space, yet their morphological similarity to the human hand holds immense potential to accelerate robot learning. We propose DexVIP, an approach to learn dexterous robotic grasping from human-object interactions present in in-the-wild YouTube videos. We do this by curating grasp images from human-object interaction videos and imposing a prior over the agent's hand pose when learning to grasp with deep reinforcement learning. A key advantage of our method is that the learned policy is able to leverage free-form in-the-wild visual data. As a result, it can easily scale to new objects, and it sidesteps the standard practice of collecting human demonstrations in a lab—a much more expensive and indirect way to capture human expertise. Through experiments on 27 objects with a 30-DoF simulated robot hand, we demonstrate that DexVIP compares favorably to existing approaches that lack a hand pose prior or rely on specialized tele-operation equipment to obtain human demonstrations, while also being faster to train.

**Keywords:** dexterous manipulation, learning from observations, learning from demonstrations, computer vision

## 1   Introduction

Many objects in everyday environments are made for human hands. Mugs have handles to grasp; the stove has dials to push and turn; a needle has a small hole through which to weave a tiny thread. In order for robots to assist people in human-centric environments, and in order for them to reach new levels of adept manipulation skill, *multi-fingered dexterous robot hands* are of great interest as a physical embodiment [1, 2, 3, 4, 5, 6, 7]. Unlike common end effectors like parallel jaw grippers or suction cups, a dexterous hand has the potential to execute complex behaviors beyond pushing, pulling, and picking, and to grasp objects with complex geometries in functionally useful ways [8, 9].

The flexibility of a dexterous robotic hand, however, comes with significant learning challenges. With articulated joints offering 24 to 30 degrees of freedom (DoF), the action space is formidable. At the same time, interacting with new objects having unfamiliar shapes demands a high level of generalization. Both factors have prompted exciting research in deep reinforcement learning (RL), where an agent dynamically updates its manipulation strategy using closed-loop feedback control with visual sensing while attempting interactions with different objects [2, 5, 7].

To mitigate sample complexity—since many exploratory hand pose trajectories will yield no reward—current methods often incorporate imitation learning [1, 2, 3, 4, 10]. With imitation, expert (human) demonstrations provided by teleoperation in virtual reality [11, 12], mocap [1, 13], or kinesthetic manipulation of the robot's body [14, 15] are used to steer the RL agent towards desirable state-action sequences. Such demonstrations can noticeably accelerate robot learning.

However, the existing paradigms for human demonstrations have inherent shortcomings. First, they require some degree of specialized setup: a motion glove for the human demonstrator to wear, a virtual reality platform matched to the target robot, a high precision hand and arm tracker, and/or physical access to the robot equipment itself. This in turn restricts demonstrations to lab environments, assumes certain expertise and resources, and entails repeated overhead to add new manipulations

5th Conference on Robot Learning (CoRL 2021), London, UK.

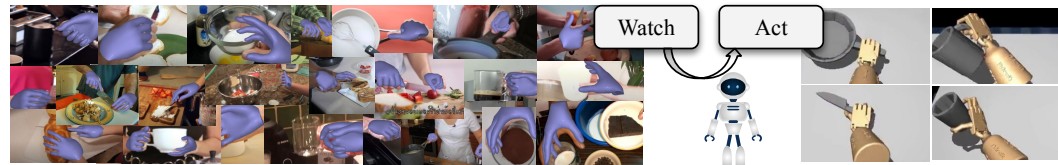

Figure 1: **Main idea**: We learn dexterous grasping by watching human-object interactions in YouTube how-to videos. Using hand poses extracted from a repository of curated human grasp images (left), we train a dexterous robotic agent to learn to grasp objects in simulation (right). The key benefits include improved grasping performance and the ability to quickly scale the method to new objects.

or objects. Second, there is an explicit layer of indirection: in conventional methods, a person does not do a task with their own hands, but instead enacts a proxy under the constraints of the hardware/software used to collect the demonstration. For example, in VR, the person needs to watch a screen to judge their success in manipulating a simulated hand and may receive limited or no force feedback [2, 16]. Similarly, advanced visual teleoperation systems that observe the bare hand [13] still separate the person's hand from the real-world object being manipulated. In a kinesthetic demonstration, the human expert is furthest removed, manually guiding the robot's end effector, which has a very different embodiment [15, 17].

In light of these challenges, we propose to learn dexterous robot grasping policies by watching people interact with objects in video (see Fig. 1). The main idea is to observe people's hand poses as they use objects in the real world in order to establish a 3D hand pose prior that a robot might attempt to match during functional grasping. Rather than enlist special purpose demonstrations (e.g., record videos at lab tabletops), we turn to in-the-wild Internet videos as the source of the visual prior. Our method automatically extracts human hand poses from video frames using a state-of-the-art computer vision technique. We then define a deep reinforcement learning model that augments a grasp success reward with a reward favoring human-like hand poses upon object contact, while also preferring to grasp the object around affordance regions predicted from an image-based model. In short, by watching video of people performing everyday activities with a variety of objects, our agents learn how to approach objects effectively using their own multi-fingered hand, while also accelerating training.

Aside from providing accurate policies and faster learning, our approach has several conceptual advantages. First, it removes the indirection discussed above. There is no awkwardness or artificiality of VR, kinesthetic manipulations, etc,; people in the videos are simply interacting with objects in the context of their real activity. This also promotes learning *functional* grasps—those that prepare the object for subsequent use—as opposed to grasps that simply lift an object in an arbitrary manner. New demonstrations are also easy to curate whenever new objects become of interest, since it is a matter of downloading additional video. In addition, because our visual model focuses on 3D hand pose, it tolerates viewpoint variation and is agnostic to the complex visual surroundings in training data (e.g., variable kitchens, clutter, etc.). Finally, the proposed method requires only visual input from the human data—no state-action sequences.

We demonstrate our approach trained with frames from YouTube how-to videos to learn grasps for a 30-DoF robot in simulation for a variety of 27 objects. The resulting policy outperforms state-of-the-art methods for learning from demonstration and visual affordances [2, 18], while also being 20% more efficient to train. The learned behavior resembles that of natural human-object interactions and offers an encouraging step in the direction of robot learning from in-the-wild Internet data.

## 2 Related Work

**Learning to grasp objects** Early grasping work explicitly reasons about an object's 3D shape against the gripper [19, 20], whereas learning methods often estimate an object or hand pose followed by model-based planning [21, 8], optionally using supervised learning on visual inputs to predict successful grasps [21, 22, 23, 24, 25]. Most planning methods cater to simple non-dexterous end-effectors like parallel jaw grippers or suction cups that make a control policy easier to codify but need not yield functional grasps. Rather than plan motions to achieve a grasp, reinforcement learning (RL) methods act in a closed loop with sensing, which has the advantage of dynamically adjusting to object conditions [26, 27, 28]. Only limited work explores RL for dexterous manipulation [5, 7, 18]. Our work addresses functional dexterous grasping with RL, but unlike the existing methods it primes agent behavior according to videos of people.

**Imitation and learning from demonstration** To improve sample complexity, imitation learning from expert demonstrations is often used, whether for non-dexterous [29, 30, 31, 32, 33, 15] or dexterous [1, 2, 3, 4] end effectors. Researchers often use demonstrations to explore dexterous manipulation in simulation [2, 3], and recent work shows the promise of sim2real transfer [11, 7]. Like learning from demonstrations (LfD), our approach aims to learn from human experts, but unlike traditional LfD, our method does so without full state-action trajectories, relying instead only on a visual prior for "good" hand states. Furthermore, our use of in-the-wild video as the source of human behavior is new and has all the advantages discussed above.

**Imitating visual observations** Learning to imitate *observations* [34] relaxes the requirement of capturing state sequences in demonstrations. This includes ideas for overcoming viewpoint differences between first and third person visual data [33, 30, 31], multi-task datasets to learn correspondences between video of people and kinesthetic trajectories of a robot arm [15], few-shot learning [29, 32], and shaping reward functions with video [33, 35, 15, 31]. However, none of the prior work uses in-the-wild video to learn dexterous grasping as we propose. By connecting real video of human-object interactions to a dexterous robotic hand, our method capitalizes on both the naturalness of the demonstrations as well as the near-shared embodiment. Furthermore, unlike our approach, the existing (almost exclusively non-dexterous) methods require paired data for the robot and person's visual state spaces [33, 15, 3], assume the demonstrator and robot share a visual environment (e.g., a lab tabletop) [29, 32, 30, 31, 3], and/or tailor the imitation for a specific object [3].

**Visual object affordances** As a dual to hand pose, priors for the object regions that afford an interaction can also influence grasping. Vision methods explore learning affordances from visual data [36, 37, 38], though they stop short of agent action. Visual affordances can successfully influence a pick and place robot [39, 40] and help grasping with simple grippers [9, 21, 23, 24] or a dexterous hand [8, 18]. Object-centric affordances predicted from images also benefit the proposed model, but they are complementary to our novel contribution—the hand-pose prior learned from video.

**Estimating hand poses** Detecting human hands and their poses is explored using a variety of visual learning methods [41, 42, 43, 44, 45]. Many methods jointly reason about the shape of the object being grasped [43, 9, 46, 47, 46]. Recent work provides large-scale datasets to better understand human hands [48, 49, 50]. We rely on the state-of-the-art FrankMocap method [44] to extract 3D hand poses from video. Our contribution is not a computer vision method to parse pose, but rather a machine learning framework to produce dexterous robot grasping behavior.

## 3 Approach

We consider the task of dexterous grasping with an articulated 30-DoF multi-fingered robotic hand. Our goal is to leverage abundant human interaction videos to provide the robot with a prior over meaningful grasp poses. To this end, we propose DEXVIP, an approach to learn **Dex**terous grasping using **V**ideo **I**nformed **P**ose priors. We first lay out the formulation of the reinforcement learning problem for dexterous grasping (Section 3.1). Then, we describe how we leverage human hand pose priors derived from in-the-wild YouTube videos for this task (Section 3.2).

### 3.1 Reinforcement Learning Framework for Dexterous Grasping

**Background** Our dexterous grasping task is structured as a reinforcement learning (RL) problem, where an agent interacts with the environment according to a policy in order to maximize a specified reward (Fig. 2). At each time step $t$, the agent observes the current observation $o_t$ and samples an action $a_t$ from its policy $\pi$. It then receives a scalar reward $R_{t+1}$ and next observation $o_{t+1}$ from the environment. This feedback loop continues until the episode terminates at $T$ time steps. The goal of the agent is to determine the optimal stochastic policy that maximizes the expected sum of rewards.

**Observations** Our task setup consists of a robotic hand positioned above a tabletop, with an object of interest resting on the table. At the start of each episode, we sample an object and randomly rotate it from its canonical orientation. The observations $o_t^r$ (Fig. 2, green block) at each time step $t$ are a combination of visual and motor inputs to the robot. The visual stream is from an egocentric hand-mounted camera. It consists of an RGB image of the scene $I_t^r$ and the corresponding depth map $D_t^r$. Additionally, we provide a binary affordance map $A_t^r$ that is inferred from $I_0^r$ using an affordance prediction network [18] to guide the agent towards functional grasp regions on the object. The motor

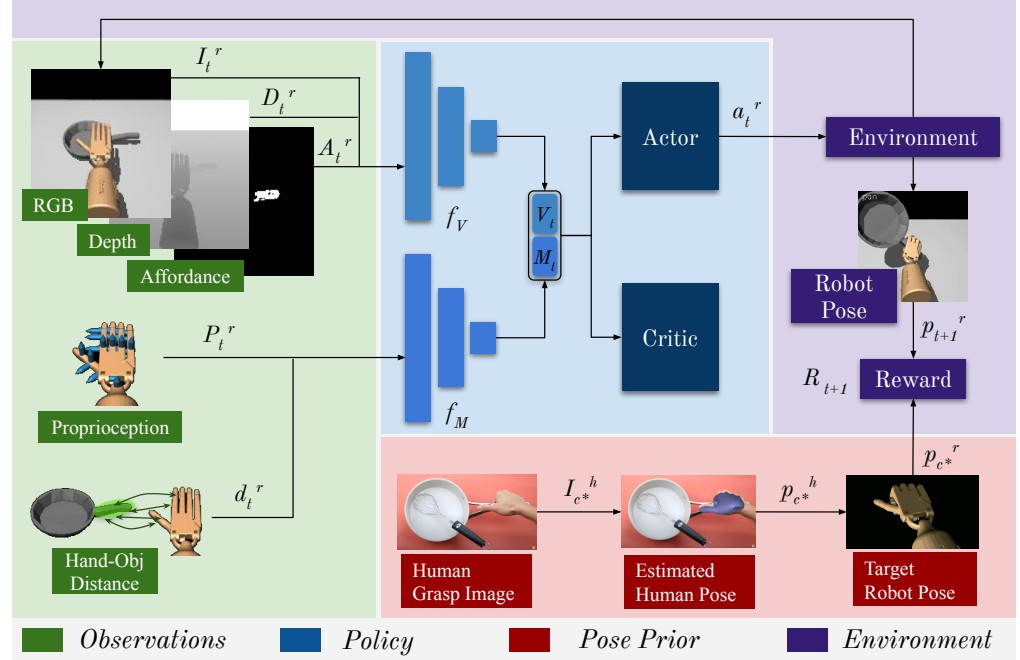

Figure 2: **Overview of DEXVIP.** We use grasp poses inferred from Internet video to train a dexterous grasping policy. An actor-critic network (blue) processes sensory observations from visual and motor streams (green) to estimate agent actions. Human hand pose priors derived from how-to videos (red) encourage the agent to explore worthwhile grasp poses via an auxiliary reward (purple).

inputs are a combination of the robot proprioception $P_t^r$ and the hand-object contact distances $d_t^r$. $P_t^r$ comprises of the robot joint angles or pose $p_t^r$ and angular velocities $v_t^r$ of the actuator, while $d_t^r$ is the pairwise distance between the object affordance regions and contact points on the hand. The latter assumes the object is tracked in 3D once its affordance region is detected, following [18]. The agent also has 21 touch sensors $T^r$ spread uniformly across the palm and fingers.

**Action space** At each time step $t$, the policy $\pi$ processes observations $o_t^r$ and estimates actions $a_t^r$—30 continuous joint angle values—which are applied at the joint angles in the actuator. The robotic manipulator we consider is the Adroit hand [51], a 30-DoF position-controlled dexterous hand. With a five-fingered 24-DoF actuator attached to a 6-DoF arm, the morphology of the robot hand closely resembles that of the human hand. This congruence opens up an exciting avenue to infuse a grasp pose prior learned from human-object interaction videos, as we present later in Sec. 3.2.

**Feature and policy learning** We adopt an actor-critic model for learning the grasping policy. The visuo-motor observations $o_t^r$ are processed separately using two neural networks, $f_V$ and $f_M$ (Fig. 2, blue block). Specifically, the visual inputs encompassing $\{I_T^r, D_T^r, A_t^r\}$ are concatenated and fed to a three-layer CNN $f_V$ that encodes them to obtain a visual embedding $V_t$. The motor stream comprised of $\{P_t^r, d_t^r\}$ is processed by a two-layer fully connected network $f_M$ that encodes them to a motor embedding $M_t$. Finally $V_t$ and $M_t$ are concatenated and fed to the actor and critic networks to estimate the policy distribution $\pi_\theta(a_t^r|o_t^r)$ and state values $V_\theta(o_t^r)$, respectively, at each time step. The resulting policy $\pi$ outputs a 30-D unit-variance Gaussian whose mean is inferred by the network; we sample from this distribution to obtain the robot's next action $a_t^r$.

We train the complete RL network with PPO [52] using a reward that encourages successful grasping, touching object affordance regions, and mimicking human hand poses, as we will detail below.

**Robot hand simulator** We conduct experiments in MuJoCo [53], a physics simulator commonly used in robotics research. Due to lack of access to a real robotic hand, we perform all experiments in simulation. The successful transfer of dexterous policies trained purely in simulation to the real world [11, 7, 6] supports the value of simulator-based learning in research today. In addition, we conduct numerous experiments with noisy sensing and actuation settings that might occur in the real world to illustrate the robustness of the policy to non-ideal scenarios (see Sec. 4 and Supp.).

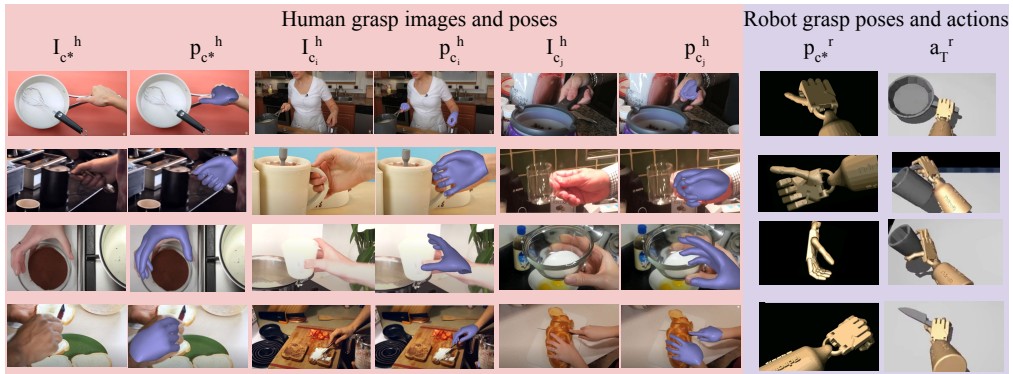

| | Human grasp images and poses | | | | | Robot grasp poses and actions | |
| $I_{c*}^h$ | $p_{c*}^h$ | $I_{c_i}^h$ | $p_{c_i}^h$ | $I_{c_j}^h$ | $p_{c_j}^h$ | $p_{c*}^r$ | $a_T^r$ |

Figure 3: **Human hand pose priors informing the agent action.** Each row shows three example images and extracted human hand poses for each object category (left) and the corresponding consensus robot hand pose rewarded by our method and its application by the agent in action (right).

### 3.2 Object-Specific Human Hand Pose Priors from Video

We now describe how we leverage human-object interaction videos to enable robust dexterous grasping. The morphological similarity between the human hand and the dexterous robot holds immense potential to learn meaningful pose priors for grasping. To facilitate this, we first curate a human-object interaction dataset of video frames to infer 3D grasp hand poses for a variety of objects. We then transfer these poses to the robotic hand using inverse kinematics, and finally develop a novel RL reward that favors human-used poses when learning to grasp. The proposed approach allows us to leverage readily available Internet data to learn robotic grasping.

**Object dataset** We consider household objects often encountered in daily life activity (and hence online instructional videos) and for which 3D models are available. We acquire 3D object models from multiple public repositories: ContactDB [37], 3DNet [54], YCB [55], Free 3D [56], and 3D Warehouse [57]. We specifically include the 16 ContactDB objects with one-hand grasps used in recent grasping work to facilitate concrete comparisons [18]. We obtain a total of 27 objects to be used for training the robotic grasping policy, all 16 from ContactDB plus 11 additional objects.

**Video frame dataset** We use the HowTo100M dataset [58] to curate images containing human-object grasps for the objects of interest. HowTo100M is a large-scale dataset consisting of 13.6 M instructional YouTube videos across categories such as cooking, entertainment, hobbies and crafts, etc. We focus on videos consisting of commonly used household objects—tools and kitchen utensils such as mug, hammer, jug, etc. The idea is to capture objects in active use during natural human interactions so that we can obtain functional hand poses. The grasp images contain the object in its canonical upright position (e.g., pan on stove), which is also the initial vertical orientation of the object on the tabletop in the simulator. Using the above criteria, we curate an object interaction repository $\mathcal{I}_h$ of 715 video frames from HowTo100M where the human hand is grasping one of the 27 total objects, to yield on average 26 grasp images per object. For instance, to collect expert data for grasping a *pan*, we curate grasp images from task ids such as "*care for nonstick pans*", "*buy cast iron pans*", etc. While we found it effective to use simple filters based on the weakly labeled categories and specific task ids in HowTo100M, the curation step could be streamlined further by deploying vision methods for detecting hands, actions, and objects in video [48, 59, 58].

**Target hand pose acquisition** We propose to use the obtained HowTo100M grasp images $\mathcal{I}_h$ to provide a learning signal for robotic grasping. To that end, for each image we first infer its 3D human hand pose $p^h$. In particular, we employ FrankMocap [44] to estimate 3D human hand poses (see Fig. 3, left). FrankMocap is a near real-time method for 3D hand and body pose estimation from monocular video; it returns 3D joint angles for detected hands in each frame. While alternative pose estimation methods could be plugged in here, we use FrankMocap in our implementation due to its efficiency and good empirical performance. We keep right hand detections only, since our robot is one-handed; we leave handling bimanual grasps for future work.

This step yields a collection of different hand poses per object for a variety of objects found in the videos. Let $\mathcal{P}(c) = \{p_{c_1}^h, \ldots, p_{c_n}^h\}$ be the set of human hand poses associated with object class $c$. The poses within an object class are often quite consistent since the videos naturally portray people

using the object in its standard functional manner, e.g., gripping a pot handle elicits the same pose for most people. However, some objects elicit a multi-modal distribution of poses (e.g., a knife held with or without an index finger outstretched). See Fig. 3. In order to automatically discover the "consensus" pose for an object, we next apply k-medoid clustering on each set $\mathcal{P}(c)$. We consider the medoid hand pose of the largest cluster to be the consensus target hand pose $p_{c*}^h$ and use its associated target robot pose $p_{c*}^r$ (obtained using a joint re-targeting mechanism described in Supp. Sec. D) during policy learning for object $c$.

**Video-informed reward function** To exert the video prior's influence in our RL formulation, we incorporate an auxiliary reward function favoring robot poses similar to the human ones in video. In this way, the reward function not only signals *where* to grasp a particular object, but also guides the agent on *how* to grasp effectively. To realize this, we combine three rewards: $R_{succ}$ (positive reward when the object is lifted off the table), $R_{aff}$ (negative reward denoting the hand-affordance contact distance obtained from [18], see Supp. Sec. F), and—most notably—$R_{pose}$, a positive reward when the agent's pose $p_t^r$ matches the target grasp pose $p_{c*}^r$ for that object. Our total reward function is:

$$R = \alpha R_{succ} + \beta R_{aff} + \gamma R_{pose} + \eta R_{entropy}, \tag{1}$$

where $\alpha, \beta, \gamma, \eta$ are scalars weighting the rewards that are set by validation, and $R_{entropy}$ rewards entropy over the target action distribution to encourage the agent to explore the action space. Through $R_{aff}$, the agent is incentivized to explore areas of the object within the affordance region, while $R_{pose}$ encourages the agent to reach hand states that are most suitable for grasping those regions.

For $R_{pose}$, we compute the mean per-joint angle error at time $t$ between the robot joints $p_t^r$ and the target grasp pose $p_{c*}^r$. We ignore the azimuth and elevation values of the arm in the pose error since they are specific to the object orientation in $I^h$, which may be different from the robot's viewpoint $I^r$. This provides more flexibility during object reaching to orient the arm while trying to match the hand pose alone. Furthermore, $R_{pose}$ is applied only when 30% of the robot's touch sensors are activated. This encourages the robot to assume the target hand pose once it is close to the object and in contact with it. Following [18], $R_{aff}$ is computed as the Chamfer distance between points on the hand and points on the inferred object affordance region. See Supp. Sec. B for reward function details.

The proposed approach permits imitation by visual observation of the human how-to videos, yet without requiring access to the state-action trajectories of the human activity. Its mode of supervision is therefore much lighter than that of conventional teleoperation or kinesthetic teaching, as we will see in results. Furthermore, because our model can incorporate priors for new objects by obtaining new training images, it scales well to add novel objects.

## 4   Experiments

We present experiments to validate the impact of learning pose priors from video and to gauge DEXVIP's performance relative to existing methods and baselines.

**Compared methods** We compare to the following methods: (**1**) COM: uses the center of mass of the object as a grasp location prior, since this location can lead to stable grasps [60]. We implement this by penalizing the hand-CoM distance for $R_{aff}$ in Eqn 1, and removing $R_{pose}$. (**2**) TOUCH: uses only the touch sensors $T^r$ on the hand to positively reward the agent $+1$ for object interaction when 30% of them are activated, but imposes no supervision on the hand pose. (**3**) GRAFF [18]: a state-of-the-art RL grasping model that trains a policy to grasp objects at inferred object-centric affordance regions. Unlike our method, GRAFF does not enforce any prior over the agent's hand pose. All the above three RL methods use the same architecture as our model for the grasping policy, allowing for a fair comparison. (**4**) DAPG [2]: is a hybrid imitation+RL model that uses motion-glove demonstrations collected from a human expert in VR. It is trained with object-specific mocap demonstrations collected by [18] for grasping ContactDB object (25 demos per object). For objects beyond ones in ContactDB, we use demos from the object most similar in shape in ContactDB.

**Metrics** We report four metrics: (**1**) Grasp Success: when the object is lifted off the table by the hand for at least the last 50 time steps (a quarter of the episode length) to allow time to reach the object and pick it up. (**2**) Grasp Stability: the firmness with which the object is held by the robot, discounting grasps in which the object can easily be dropped. We apply perturbation forces of 1 Newton in six orthogonal directions on the object after an episode completes. If the object continues to be grasped

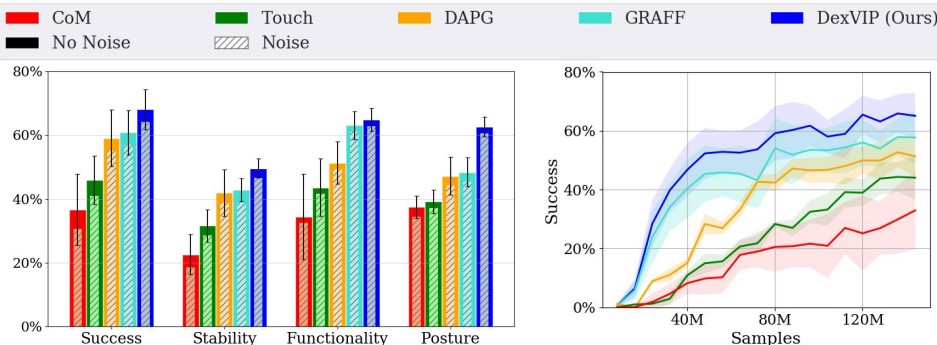

Figure 5: **Grasping success and learning speed.** Left: The proposed model outperforms all the baselines, including recent methods for learning from demonstrations (DAPG [2]) or favoring visual affordance regions (GRAFF [18]). Relative results remain stable under noisy sensing and actuation (shaded bars). Right: Our model trains faster than the others, showing that the Internet videos of people help kickstart learning even before the agent begins attempting its own grasps.

by the agent, the grasp is deemed stable. (**3**) Functionality: the percentage of successful grasps in which the hand lies close to the GT affordance region. This metric evaluates the utility of the grasp for post-grasp functional use. (**4**) Posture: the distance between the target human hand pose $p_{c*}^h$ and the agent hand pose after a successful grasp $p_T^r$. It tells us how human-like the learned grasps are. We normalize all metrics on a $[0, 100\%]$ scale, where higher is better. We evaluate 100 episodes per object with the objects placed at different initial orientations ranging from $[0, 180°]$. We report the mean and standard deviation of the metrics across all models trained with four random seeds.

**Implementation details** The visual encoder $f_V$ has filters of size [8,4,3], and a bottleneck 512-D and uses ReLU activations. The motor encoder $f_M$ has dimensions [512,512]. For the hand-object contacts, we use 10 and 20 uniformly sampled points on the hand and affordance region, respectively, following [18]. The entire network is optimized using Adam with a learning rate of $5e - 5$. A single grasping policy is trained on all the curated objects for 150M agent steps with an episode length of 200 time steps. The coefficients in the reward function (Eq. 1) are set as: $\alpha = 1, \beta = 1, \gamma = 1, \eta = 0.001$. We train for four random seed initializations. Further details are provided in Supp. Sec. E.1.

**Grasping policy performance** We take the policy trained on all 27 objects and first evaluate it on the 16 objects from ContactDB [37]. Since these objects have ground truth affordances (used when training GRAFF and our model) and mocap demonstrations (used when training DAPG), they represent the best case scenario for the existing models to train with clean expert data. Note that our method always uses hand poses inferred from YouTube videos, even for the ContactDB objects.

Fig 5 (left) shows the results. DEXVIP consistently outperforms all the methods on all metrics. The grasp success and stability rates experience a significant boost even compared to GRAFF [18], which utilizes object affordances but does not enforce any constraints on the hand pose. The Functionality values are similar as both the methods encourage the agent to grasp the object at the affordance regions. Our method also scores well on the Posture metric, indicating that the learned policies indeed demonstrate human-like behavior during grasping. See Supp. Sec. G and Sec. E.2 for additional results and Sec. C for TSNE plots illustrating our model's human-like poses.

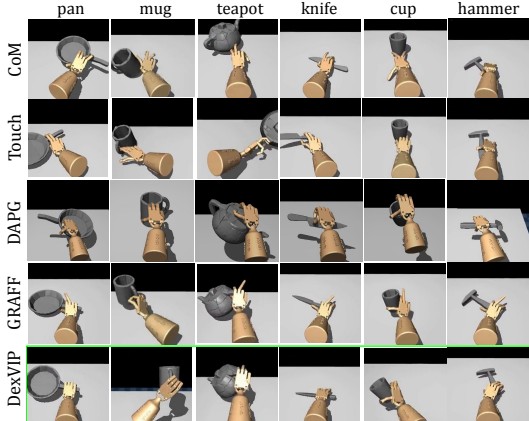

Figure 4: **Grasping performance.** Example frames for the grasping task. Our DEXVIP policy guided by pose-priors is able to successfully grasp objects in natural human-like poses, while the other methods may either generate unusual poses or fail to grasp effectively. Please see Supp. video.

Fig. 4 shows grasp policies from sample episodes. We can see that DEXVIP is able to grasp objects with a human-like natural posture compared to the other methods. Contrast this with CoM or GRAFF, which only uses object affordances: those agents often end up in un-

usual poses which reduce their applicability to post-grasp functional tasks using the object. Failure cases arise when some objects are in orientations not amenable to the target hand pose. For instance, a knife with the handle on the left would ideally be picked up differently and then re-oriented for use.

**Training speed** Fig. 5 (right) shows the training curves. Our method learns successful policies $20\%$ faster than the next best method, GRAFF [18]. This underscores how the hand pose priors enable our agent to approach objects in easily graspable configurations, thus improving sample efficiency.

**Expert data: teleoperation vs. YouTube video** Compared to traditional demonstrations, a key advantage of our video-based approach is the ease of data collection and scalability to new objects. We analyze the time taken to collect demonstrations for DAPG [2] versus the time taken to curate videos for DEXVIP. On average, it takes 5 minutes to collect a single demonstration for DAPG owing to the complex setup, while a video or image for DEXVIP is collected in a few seconds. Fig. 6 quantifies the impact of the efficiency of human demonstrations for our model (trained with video frames) compared to traditional state-action demonstrations (trained with VR demos). Plotting success rates as a function of accumulated demo experience, we see how quickly the proposed image-based supervision translates to grasping success on an increasing number of new objects, whereas with traditional demonstrations reaching peak performance takes much longer. This highlights the significant gains that can be realized by shifting from tele-op to video supervision for robot learning.

This efficiency also means new objects are quick to add. To illustrate, we further evaluate DEXVIP and DAPG on all 11 non-ContactDB objects, for which mocap demonstrations are not available. DAPG's success rate drops from $59\%$ to $50\%$—a $15\%$ relative drop in performance—while DEXVIP experiences a marginal $4\%$ drop from $68\%$ to $65\%$, remaining comparable to its performance on ContactDB. Since DAPG is trained on sub-optimal demonstrations, this precludes it from generalizing well to objects for which expert data is absent. DEXVIP, on the other hand, is able to benefit from easily available Internet images to scale up supervision.

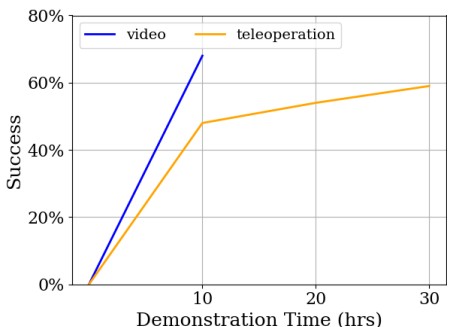

Figure 6: **Demo-time vs success rate.** DexVIP benefits from easily available Internet data to scale up supervision.

**Ablations** Next we investigate how different components of the reward influence the dexterous grasping policy. Using only $R_{aff}$, the success rate is $60\%$ on ContactDB. When we successively add touch and hand pose priors to the reward function (Eq. 1), we obtain success rates of $63\%$ and $68\%$ respectively. Thus our full model is the most effective.

**Noisy sensing and actuation** Finally, to mimic non-ideal real-world scenarios, we induce multiple sources of noise into our agent's sensory and actuation modules during training and testing following prior work [11, 7, 6, 18]. These include Gaussian noise on the robot's proprioception $P_t^r$, object tracking $d_t^r$, and actuation $a_t^r$, as well as pixel perturbations on the image observations $I_t^r$. Under heavy noise, DEXVIP still yields a grasp success rate of $64\%$, even outperforming noise-free models of the other methods (Fig. 5, shaded bars). This encouraging result in a noise-induced simulation environment lends support for potentially transferring the learned policies to the real world [6, 7, 61] were we to gain access to a dexterous robot. Please see Supp. Sec. A for more details.

## 5   Conclusion

We proposed an approach to learn dexterous robotic grasping from human hand pose priors derived from video. By leveraging human-object interactions in YouTube videos, we showed that our dexterous grasping policies outperformed methods that did not have access to these priors, including two state-of-the-art models for RL-based grasping. Key advantages of our approach are 1) humans are observed directly doing real object interactions, without the interference of conventional demonstration tools; and 2) expert information from video sources scales well with new objects. This is an encouraging step towards training robotic manipulation agents from weakly supervised and easily scalable in-the-wild expert data available on the Internet. In the future, we are interested in expanding the repertoire of tasks beyond grasping to learn fine-grained manipulation skills from human interaction videos.

**Acknowledgments**

UT Austin is supported by DARPA L2M, the IFML NSF AI Institute, and NSF IIS -1514118. K.G. is paid as a research scientist at Facebook AI Research. We would like to thank the reviewers for their valuable feedback.

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
