# OpenReview forum: "DexVIP: Learning Dexterous Grasping with Human Hand Pose Priors from Video"
_robot-learning.org/CoRL/2021/Conference — CoRL2021 Poster_

### Official Review · Reviewer_HJKw · 2021-07-20

**Originality:** Good
**Technical Quality:** Good
**Clarity Of Presentation:** Very Good
**Impact:** 3

**Recommendation:**

Weak Accept: I recommend accepting the paper, but will not argue for my recommendation if the majority of other reviewers have a different opinion.

**Summary:**

This paper presents DexVIP, a method for learning a policy for dexterous grasping. Grasping with a dexterous, multi-fingered robot hand is a challenging problem; while the hand’s high degrees of freedom (DOF) make it possible to perform behaviors that are beyond simple parallel-jaw grippers, the large action space presents significant learning challenges. The key contribution of DexVIP is to use human hand pose priors when training the grasping policy, to improve sample complexity and produce human-like grasps. Hand pose priors are first extracted using FrankMocap [1] from in-the-wild video demonstrations of humans grasping an object similar to the desired object. By using video demonstrations, DexVIP does not require expensive or difficult-to-use hardware for collecting demonstrations. The human hand pose priors are incorporated as the “R_pose” term in the reward function (Eq. 1) to encourage the robot hand to use the same grasp pose as the human demonstrator. The method is otherwise identical to GRAFF [2], a state-of-the-art dexterous grasping approach. The paper presents experiments with a simulated 30-DOF hand evaluating DexVIP against a number of baselines: GRAFF, a hybrid imitation+RL approach named DAPG [3], and alternative reward terms such as grasping towards the object center of mass and rewarding contact with the object. The paper also provides additional experiments comparing video-based vs. teleoperation-based demonstrations, ablations, and the effect of training with noise.

**Issues:**

- Run an ablation removing the object affordance reward R_aff
- Evaluate all methods only on the 11 objects not in ContactDB
- Evaluate all methods only on object classes not in ContactDB
- Clarify how object affordance maps are obtained for non-contact DB objects
- Clarify the difference between the state space between the human and robot hands


**Reviewer Expertise:**

Good: General knowledge of the area

**Strengths And Weaknesses:**

Strengths
- The proposed method helps pave the way towards dexterous robot manipulation
- The data used for training the pose prior can be collected from in-the-wild videos and is hence easy to collect and easily scalable. This data involves people directly interacting with objects, as opposed to the more indirect demonstration approaches of VR, teleoperation, or kinesthetic teaching.
- The proposed method is somewhat invariant to viewpoint variation, since the reward focuses on the pose of the individual hand joints (regardless of the position / orientation of the hand itself)
- The paper evaluates DexVIP against existing methods and shows that DexVIP outperforms them on all evaluation metrics. Specifically, DexVIP slightly outperforms other methods on grasp success (% grasps that lifted the object for more than 50 timesteps), stability (% grasps robust to perturbations), and functionality (% grasps close to a ground-truth affordance region of the object). DexVIP achieves significant outperformance on posture (normalized distance of robot grasp pose to human grasp pose), a metric its additional reward term is designed to maximize.
- DexVIP achieves better performance with less training time (Fig 5, right) and less demonstration time (Fig 6) compared to the alternative methods.
- The paper provides ablation experiments demonstrating the importance of the hand pose reward term, as well as experiments showing that DexVIP is robust to observation and actuation noise during training.
- The paper is well-structured and written clearly.

Major Weaknesses:

**Scalability is limited by ContactDB data:**
The paper argues in lines 49-59, 65-66, and 303-326 that DexVIP scales better to new objects because of its use of video demonstrations for extracting hand pose priors (“New demonstrations are also easy to curate whenever new objects become of interest, since it is a matter of downloading additional video.”). However, the reward used for DexVIP still requires R_aff, the object affordance regions. If the object affordance maps require expensive human demonstrations with thermal cameras (on 3D-printed objects) for each object or object class, then this argument does not hold; the scalability would still be limited by the ability to collect this affordance data. The following additional experiments should be performed to measure the true scalability of DexVIP:

1) Train DexVIP without the affordance reward term and show the performance (this would show how scalable the method is based only on Youtube data)

2) Assuming that DexVIP makes use of the “Image affordance prediction model” from GRAFF (and this should be explicitly stated in the DexVIP paper), experiments should be added to show the performance of DexVIP on unseen object instances and unseen object classes for which ContactDB data has not been obtained. These experiments would show that additional thermal ContactDB data is not added for new instances or classes, and that the proposed method can rely on the generalization of the affordance prediction, as well as the scalability of Youtube videos. For the novel object instances, I believe that this can be achieved by training and evaluating only on the 11 non-ContactDB objects.This experiment is nearly done in the section “Expert data: teleoperation vs. YouTube video”, but that experiment only compares the performance of DexVIP and DAPG instead of comparing all methods). For novel object classes, an additional experiment should be added to show the scalability of DexVIP to novel classes for which ContactDB data is not available.

**Limited technical novelty:**
DexVIP has limited technical novelty, as it is mostly a combination of existing methods: GRAFF + FrankMocap.

**Minimal performance improvement:**
The results show only a small improvement over GRAFF, especially in grasping success, stability, and functionality, which are the most important metrics.  The ablation experiments show
that adding the hand pose prior increases grasp success rate from 63 to 68%, a relatively small increase. There is a large improvement in “posture”, but this metric is the least important because it doesn’t really matter whether the robot performs a task in a human-like way, as long as the task is successfully completed.



Minor Weaknesses
- Lines 271-272 state: “We take the policy trained on all 27 objects and first evaluate it on the 16 objects from ContactDB.” This seems like an odd procedure; it seems that a more straightforward evaluation would be to just train and evaluate on the 16 objects from ContactDB rather than training on 27 objects and only evaluating on a subset of them.
- Is the state space between the human and robot hands the same?  This isn’t discussed; if not, how do they bridge the morphological gap for computing the R_pose reward term? It is somewhat surprising to me that FrankMocap and the Adroit hand happen to use the same state space.
- DexVIP requires both hand pose priors extracted from video for its R_pose metric, as well as object affordance maps for the object affordances metric R_aff. For the 16 ContactDB training objects, the ground truth object affordance maps used in training are obtained from human demonstrations using thermal cameras. For non-ContactDB training objects, it is unclear from the paper how the object affordance maps are obtained. It is crucial to explain how these object affordance maps are obtained.
- Fig. 6: The caption is not very descriptive compared to the other figure captions.
- Supp. line 20: For the details in the supplement on the experiment with noisy observations and actions, is the 0.1 variance value in radians? This value does not seem to be used in [4] or [5] for similar noise-injected experiments.
- Are the objects in the in-the-wild video demonstrations and the object meshes matched manually? This could be clarified in the paper.
- For the analysis in Appendix C, it would be interesting to see how often grasps that are similar according to standard taxonomies (https://www.is.mpg.de/uploads_file/attachment/attachment/256/grasp_taxonomy.pdf) are grouped together.

[1] Y. Rong, T. Shiratori, and H. Joo. Frankmocap: Fast monocular 3d hand and body motion capture by regression and integration. arXiv preprint arXiv:2008.08324, 2020.

[2] P. Mandikal and K. Grauman. Dexterous robotic grasping with object-centric visual affordances. In arXiv preprint arXiv:2009.01439, 2020.

[3] A. Rajeswaran, V. Kumar, A. Gupta, G. Vezzani, J. Schulman, E. Todorov, and S. Levine. Learning complex dexterous manipulation with deep reinforcement learning and demonstrations. Robotics: Science and Systems (RSS), 2018.

[4] Y. Zhu, Z. Wang, J. Merel, A. Rusu, T. Erez, S. Cabi, S. Tunyasuvunakool, J. Kram r, R. Hadsell, N. de Freitas, et al. Reinforcement and imitation learning for diverse visuomotor skills. arXiv preprint arXiv:1802.09564, 2018.

[5] OpenAI, M. Andrychowicz, B. Baker, M. Chociej, R. Jozefowicz, B. McGrew, J. Pachocki, A. Petron, M. Plappert, G. Powell, A. Ray, et al. Learning dexterous in-hand manipulation. The International Journal of Robotics Research, 2020.


**Summary Of Recommendation:**

Extracting hand pose priors from in-the-wild video demonstrations seems promising for learning dexterous grasping with human-like hand poses. However, the current version of the approach seems to rely on other types of demonstrations such as thermal human demonstrations on 3D-printed objects for collecting ground truth object affordance maps, which weakens the claim made in the paper that DexVIP scales better for learning new objects. Further, the technical novelty is limited and the performance improvement compared to prior work is minimal.

# Post-rebuttal update

I would like to clarify that my concerns about the evaluation are not based on a misunderstanding.  The issue is that the 16 objects that were used in the main evaluation in the paper (Figure 5) were in the *training set* for the affordance prediction model.  Because these objects were in the training set, this evaluation does not demonstrate the generalization of the affordance prediction model; contactDB data was acquired and used for these specific objects that were used in the evaluation of Figure 5. The use of contactDB data significantly limits the scalability of the method, and it is unclear from Figure 5 whether the proposed method would still outperform the baselines on objects for which contactDB data is not available.  This is an issue because Figure 5 is the only figure in the main text in which DexVIP is compared to all of the baselines, and it is the only figure in the main text in which DexVIP is evaluated on a full range of metrics.

This issue is mostly rectified in Supp Sec. G.4 and Figure F. I would recommend to replace Figure 5 with Figure F in the final version of the paper. The downside of the approach used in Figure F, though, is that the functionality metric can no longer be included. Ideally, the paper should evaluate DexVIP on ContactDB objects that are in the affordance prediction model's test set (objects not used for affordance prediction training) so that the functionality metric can also be included, while still demonstrating generalizability to objects beyond those in the affordance prediction model's training set.

One small typo: the main text refers to "Supp Sec H", but the sections in the supplement only goes up to G.

Overall, I think the authors have significantly improved the paper during the rebuttal and addressed most of my concerns; with these updates, I think that the paper represents a solid contribution and thus I will update my recommendation to weak accept.

---

> ### Author Response · Authors · 2021-08-28
> **Author Response (1/2)**
>
> Thank you for your detailed review and comments. We think a misunderstanding about how the thermal images from ContactDB are used and how affordance regions are predicted may have affected the reviewer’s overall opinion on our paper. We clarify all items below.
>
> ### 1. “Requires thermal images for each object, so scalability limited”
> We do not require thermal images for every object in the dataset, as affordances are inferred from the anticipation model from GRAFF [18], as stated in L137-138. The affordance anticipation model, although trained on thermal maps from ContactDB, is able to generalize well to new objects as shown in GRAFF. This is corroborated by both GRAFF as well as our model. So we do not assume thermal image availability for all objects and DexVIP scales well to objects beyond just ones in ContactDB (L314-325 main paper and Supp Sec. G.4).
>
> ### 2. “Train DexVIP without the affordance reward”
> As noted in L278-282, grasp success and stability rates experience a significant boost compared to GRAFF. Our DexVIP method is complementary to GRAFF and imposes priors on the robot’s hand while approaching an object at its affordance region. While GRAFF tells the agent where to grasp the object, DexVIP provides a signal for how to grasp it.
> Per the reviewer’s request, for a model trained without affordances, the metrics are reported in the table below for both ContactDB and non-ContactDB objects:
>
> | Model    | Inputs | Success | Stability |
> |-------------|----------|---------|-----------|
> | GRAFF   |   RGB + Depth + Obj Aff + Proprioception         	| 60 / 53   | 42 / 37   |
> | DexVIP - no R_aff  |   RGB + Depth + Proprioception       	| 64 / 60   | 46 / 42  |
> | DexVIP - full  |   RGB + Depth + Obj Aff + Proprioception   | 68 / 65   | 50 / 45  |
> ||
>
> \* the metrics are reported as:  ContactDB / non-ContactDB
>
> Thus our model improves over the GRAFF results even without using the object affordance regions, and the improvement is even larger for non-ContactDB objects. The full DexVIP model sees substantial gains compared to GRAFF on all metrics.
>
> ### 3. Performance on Non-ContactDB objects
> In addition to comparing performance on non ContactDB objects with DAPG (L314-325), we provide a comparison against all methods in Fig. F in Supp. Note that all 11 non-ContactDB objects belong to object classes not found in ContactDB. When compared to performance on ContactDB objects, we observe that DexVIP experiences only a marginal drop, whereas the other methods suffer substantial drops in performance. As reported in L318-320, DAPG’s success rate drops from 59%-50%, a 15% relative drop in performance, while DexVIP sees a marginal 4% drop from 68% to 65%. Furthermore GRAFF also sees a large 12% drop from 60% to 53%. The results indicate that DexVIP can effectively leverage hand poses for a variety of objects---even those which have no thermal images.
>
> ### 4. Technical novelty
> Our method is the first to learn a dexterous manipulation policy directly from in-the-wild YouTube data. We believe that this is an important contribution towards building robotic systems that can leverage vast internet data instead of depending on complex tele-operation setups [2] to tackle the high-dimensional state space of a dexterous hand. We demonstrate substantial improvement in grasping performance over existing methods, while being efficient and easy to scale. Furthermore, we also provide a technique to transform the pose from the human state space to the robot’s state space in the simulator. This is a step towards building efficient, scalable and general-purpose robot learning systems.
>
> ### 5. Improvements
> DexVIP has a healthy 13% relative improvement over GRAFF using touch-based pose priors  The pose priors themselves contribute to an 8% improvement over an affordance+touch model. Considering that the improvement from DAPG [2] to GRAFF [18] (two state-of-the-art methods) is a 1% improvement in success rate, we believe the gains made from GRAFF to DexVIP are substantial and meaningful across all metrics. As can be seen in Supp Fig. F, the improvement over GRAFF for non-ContactDB objects is even higher with a 23% gain in success rate.
>
> **Continued below...**

---

> > ### Author Response · Authors · 2021-08-28
> > **Author Response (2/2)**
> >
> > **...continued from Response (1/2) above**
> >
> > ### 6. “Doesn’t matter if done in human-like way”
> > We tackle the problem of functional grasping for object use, instead of arbitrary pick and place. The posture metric is a useful indicator of functional grasping since the target poses are obtained from how-to human-object interaction videos involving post-grasp functional use. Picking up objects in a more human-like way not only improves performance (as indicated by the success metric), but also has a significant impact on the grasp stability. This is especially important when the agent is grasping objects for subsequent use in home and kitchen environments. We observe that GRAFF often picks up objects using a single finger, while DexVIP is able to execute fully clasped grasps (please see Supp video). While the former might be sufficient to just pick up and drop an object, the latter is crucial for using the object for post-grasp functions like pouring, stirring, etc.
> >
> > ### 7. Minor Weaknesses
> > 1. Policy training: We train a single policy on all objects so that it is easier to scale the number of experiments and perform ablations. The conclusion from the results obtained remains the same regardless of whether we train on 27 or 16 objects.
> > 2. Hand Pose Retargeting from FrankMocap to Adroit: Please refer to Vrhm-1 for an overview. Full details have been added to  Supp Section D and Fig. B.
> > 3. Affordance maps: The affordance maps are obtained from the affordance prediction model from GRAFF (L137-138). This model is able to anticipate affordance maps for novel objects not present in ContactDB. We have made this more explicit in the main paper (L218).
> > 4. Caption for Fig 6: Thanks, we have modified this.
> > 5. Noise model for proprioception: This is indeed a typo, thanks for pointing it out! Following [4] (Zhu et al. 2018), we apply a Gaussian noise of mean 0 and standard deviation 0.01 in our experiments. This has been corrected in Supp L30-31.
> > 6. Object alignment: The object labels are obtained automatically using meta-data from the HowTo100M videos. So manual curation of the object type is not required as explained in Vrhm.VI.3. As stated above and in L190-192 of the main paper, since we don’t require an exact match of the object instance (Supp Sec. G.2), such an automatic curation of object images is effective. While many existing works in robotic manipulation [2, 3, 8] assume a direct alignment between the demonstration objects and simulator objects, we consider a weaker alignment of object *class* but not object *instance*. Future work can look into image-based shape matching to detect the right model in the absence of class labels.
> > 7. Grasp taxonomy: The supplementary video shows data points along the T-SNE plot for both the human poses as well as the robot pose, along with grasp types from a standard taxonomy. The T-SNE plots indicate that there is a gradual shift from outstretched fingers (for objects like ball, scissors, etc) to clenched fists (for pan, hammer, etc). From a qualitative observation, we notice that most grasps fall under the power grasp category. While the outstretched finger grasps could be classified as a pad grasp, the clenched fist grasps fall under the palmar grasp category. This indicates an alignment with the grasp taxonomy in https://www.is.mpg.de/uploads_file/attachment/attachment/256/grasp_taxonomy.pdf, where objects like ball and scissors fall under the pad grasp, whereas cylinders (similar to object handles) fall under palm grasps.

---

> > > ### Comment · Reviewer_HJKw · 2021-08-29
> > > **Reviewer response**
> > >
> > > > We do not require thermal images for every object in the dataset, as affordances are inferred from the anticipation model from GRAFF [18], as stated in L137-138. The affordance anticipation model, although trained on thermal maps from ContactDB, is able to generalize well to new objects as shown in GRAFF. This is corroborated by both GRAFF as well as our model. So we do not assume thermal image availability for all objects and DexVIP scales well to objects beyond just ones in ContactDB (L314-325 main paper and Supp Sec. G.4).
> > >
> > > I am referring to lines 271-274: "We take the policy trained on all 27 objects and first evaluate it on the 16 objects from ContactDB [37]. Since these objects have ground truth affordances (used when training GRAFF and our model)..."
> > >
> > > **Were these 16 objects were in the affordance prediction model's training set or test set?**  If they were in the affordance prediction model's training set, then these experiments (specifically, Figure 5) would not demonstrate any degree of generalization of affordance prediction to novel objects for which contactDB data is not available.  Because Figure 5 is the main figure in the paper in which DexVIP is compared to all of the baselines on a range of metrics, this is a major limitation in the results.
> > >
> > > I see that this has been rectified in Supp Sec. G.4 .  If the 16 objects used in Figure 5 were in the affordance prediction model's training set, then I would recommend to replace Figure 5 with Figure F in the final version of the paper. The downside of Figure F, though, is that the functionality metric can no longer be included. I suppose the ideal would be to evaluate DexVIP on ContactDB objects in the affordance prediction model's test set (objects not used for affordance prediction training) so that the functionality metric can be included.

---

> > > > ### Author Response · Authors · 2021-08-30
> > > > **Author Response**
> > > >
> > > > **"Were these 16 objects in the affordance prediction model's training set or test set?"**
> > > > We have the 3D affordance region GT from ContactDB, so that allows scoring functionality, but even for these objects, we use the image-based affordance model to infer the affordance region, even for the 16 ContactDB objects. The latter is necessary because our input to the affordance module is a photo of the object at a novel orientation/scale. Our point in L271-274 was that for the 16 ContactDB objects, the affordance model has the best case scenario of having seen those objects during training. However, we are not providing a ground truth thermal map at test time.
> > > >
> > > >
> > > > **"I would recommend to replace Figure 5 with Figure F in the final version of the paper"**
> > > > We are glad that the performance reported for additional objects across all methods in Fig. F brings greater clarity to the results presented. In the original submission, we reported results for additional objects in comparison to DAPG [2] (L314-325) in addition to a cost analysis (Fig. 6). We agree with the reviewer that Fig. F brings more clarity, and we will make this clear in the paper.

---

> > > > > ### Comment · Reviewer_HJKw · 2021-09-03
> > > > > **Post-rebuttal update**
> > > > >
> > > > > (copying from my review update)
> > > > >
> > > > > I would like to clarify that my concerns about the evaluation are not based on a misunderstanding. The issue is that the 16 objects that were used in the main evaluation in the paper (Figure 5) were in the training set for the affordance prediction model. Because these objects were in the training set, this evaluation does not demonstrate the generalization of the affordance prediction model; contactDB data was acquired and used for these specific objects that were used in the evaluation of Figure 5. The use of contactDB data significantly limits the scalability of the method, and it is unclear from Figure 5 whether the proposed method would still outperform the baselines on objects for which contactDB data is not available. This is an issue because Figure 5 is the only figure in the main text in which DexVIP is compared to all of the baselines, and it is the only figure in the main text in which DexVIP is evaluated on a full range of metrics.
> > > > >
> > > > > This issue is mostly rectified in Supp Sec. G.4 and Figure F. I would recommend to replace Figure 5 with Figure F in the final version of the paper. The downside of the approach used in Figure F, though, is that the functionality metric can no longer be included. Ideally, the paper should evaluate DexVIP on ContactDB objects that are in the affordance prediction model's test set (objects not used for affordance prediction training) so that the functionality metric can also be included, while still demonstrating generalizability to objects beyond those in the affordance prediction model's training set.
> > > > >
> > > > > One small typo: the main text refers to "Supp Sec H", but the sections in the supplement only goes up to G.
> > > > >
> > > > > Overall, I think the authors have significantly improved the paper during the rebuttal and addressed most of my concerns; with these updates, I think that the paper represents a solid contribution and thus I will update my recommendation to weak accept.

---

### Official Review · Reviewer_Vrhm · 2021-07-23

**Originality:** Good
**Technical Quality:** Good
**Clarity Of Presentation:** Very Good
**Impact:** 3

**Recommendation:**

Weak Accept: I recommend accepting the paper, but will not argue for my recommendation if the majority of other reviewers have a different opinion.

**Summary:**

This work proposes an approach to learn dexterous robotic grasping from human-object interactions recorded in the video. Target hand poses are predicted from the video leveraging an existing method. The distances between target hand poses and current hand poses are used as a reward signal to train a deep reinforcement learning algorithm in the simulation. Besides the target hand poses, the affordance of each object is pre-calculated. The distances between the hand and affordance regions are also used as a reward signal. The proposed pipeline enables a 30-Dof hand to grasp 27 objects in the simulation achieving 68% success rates.

**Issues:**

1) In Line 137, "Additionally, we provide a binary affordance map At that is inferred from I0 using an affordance", Is the affordance map inferred from "I_0" or "I_t"
2) In Line 141, "is the pairwise distance between the object affordance regions and contact points on the hand". How are these affordance regions defined? How to calculate the distance between a region on the object with a contact point on the hand? Are the number of regions varying between different objects?
3) In Line 189-190, "we curate an object interaction repository Ih of 715 video frames from HowTo100M where the human hand is grasping one of the 27 total objects, to yield on average 26 grasp images per object." The object in the video and the object for the simulation are not exactly identical. How to make sure that the object in the video and the object for the simulation are similar enough? Are they verified automatically or by human experts?
4) In Line 197-200, "In particular, we employ FrankMocap [44] to estimate 3D human hand poses (see Fig. 3, left). FrankMocap is a near real-time method for 3D hand and body pose estimation from monocular video; it returns 3D joint angles for detected hands in each frame"
In FrankMocap[40], the 3D joints angles are based on the SMPL-X hand model (Expressive body capture: 3d hands, face,
and body from a single image. Pavlakos et al., CVPR 2019).  How to make sure that the designs and joints of SMPL-X hand model are similar to the hand used in the Mujoco simulation? There are 20 joints in SMPL-X hand while there are 24 joints in the hand used in this work. How to calculate the similarity between the poses of two hands?
5) In Line 209-210, "we next apply k-medoid clustering on each set P(c). We consider the medoid hand pose of the largest cluster to be the consensus target hand pose" What metric space is used for applying k-medoid clustering here?
6) In Line 224-225, "For Rpose, we compute the mean per-joint angle error at time t between the robot joints prt and the target grasp pose". A similar question is in Issue 4).  The two hands have different joint designs. How to measure their error?
7) In Ablation, how do the prediction errors of the target pose affect the learned policy? how does the similarity between the object in the video and the object needed grasping affect the learned policy?
8) In Ablation, why only success rates are reported? How about the rest three metrics including stability, functionality and posture?



**Reviewer Expertise:**

Very good: Comprehensive knowledge of the area

**Strengths And Weaknesses:**

Strengths:
1) This work proposes to leverage the people's hand poses from the video to train dexterous robotic grasping policy.
2) Extensive comparison experiments are conducted to demonstrate the effectiveness of the proposed approaches.
3) The paper is well-organized and easy to follow.

Weaknesses:
1) Some important information is missing. a) how to calculate the difference between the target hand poses and the current hand poses since the target hand poses returned from the FrankMocap are based on a different hand.  b) how to determine whether there are objects in the video dataset which are similar to the objects that needed grasping.
2) The sim2 real gap remains unclear. What mass values are assigned to these objects? What friction coefficients are used? Are these values sensitive in the learned policy?
3) The comparison experiments on functionality and posture are kind of unfair. Because these attributes are used during the training process. For example, the proposed method is optimized to grasp under similar postures. Of course, it achieves the highest score in the posture metric.
4) It's unclear that how the prediction errors of the target pose affect the learned policy and how the similarity between the object in the video and the object to be grasped affects the learned policy.


**Summary Of Recommendation:**

This work proposes to leverage the people's hand poses from the video to train dexterous robotic grasping policy. The proposed pipeline enables a 30 DoF hand to grasp 27 objects in the simulation with 68% grasp success rates. However, some important explanations of the proposed method are missing. It's unclear how the errors in the human pose prior affect the grasping performances. I would be happy to increase the evaluation score if these concerns are addressed.

---

> ### Author Response · Authors · 2021-08-28
> **Author Response (1/2)**
>
> Thank you for the detailed review and for pointing out missing details. We have added supporting details in the supplementary for the joint pose retargeting mechanism (Section D), simulator parameter settings and robustness analysis (Section E), hand-affordance regions (Section F, augmenting Section B in the submission), effect of pose prediction on performance (Section G.1) and full ablation metrics (Section G.2, augmenting L326-330 main paper). Please find details to specific queries below.
>
> ### 1. Hand Pose Retargeting from FrankMocap to Adroit
> Complete details about the retargeting mechanism have been added to Supp Section D and Fig. B. Here we briefly describe the retargeting procedure.
> We first obtain the FrankMocap pose i.e. 3D joint locations in the world coordinate frame. This is converted to a root relative coordinate frame through a simple coordinate translation. We then compute the palmar plane in FrankMocap to obtain the arm orientation for Adroit.  Subsequently, the structure of the kinematic tree is used to successively transform the root relative coordinate frame to a parent relative frame centered at each joint. The polar coordinates (azimuth and elevations) computed in the parent relative system yield local joint angles that are mapped onto the revolute joints in Adroit.
> Using the above re-targeting scheme, we are able to successfully transfer the FrankMocap pose from the human pose space to the Adroit pose space. Samples can be seen in Fig. 3, main paper. While the mapping is approximate, due to the inherent differences in kinematic chains as discussed above, we find that this re-targeting mechanism generates Adroit poses that closely match the human pose and works well for our purpose.
> For computing the distance between this target hand pose from the current pose of the robot for R_pose (L224-225, main paper), we use the reward function defined in Supp Sec B, Eq 2 in the original submission.
>
> ### 2. Similarity of objects in video dataset
> We use keywords containing the object class label (L190-192, main paper) for obtaining grasp images for each object. We use the same object category in simulation as well. Note that we do not require an exact matching object instance for the one in the video. A generic object mesh from the same category works quite well. To illustrate this, we show samples of a few objects in the video frame and within the simulator along with their grasp success rate in Fig. D in Supp. We find that DexVIP remains fairly robust to variations in the object shape between the video and simulator. For instance, even though objects like teapot, flashlight and saucepan don't have an exact match in the video, the grasp policy works quite well on these objects. We have added this detail in Supp Sec G.2.
>
> ### 3. Simulator settings
> We use a mass value of 1 kg for all objects during training and testing. The contact friction parameters for dynamically generated contact pairs are set to (1, 0.5, 0.01) N as in [2]. The first number is the sliding friction, acting along both axes of the tangent plane. The second number is the torsional friction, acting around the contact normal. The third number is the rolling friction, acting around both axes of the tangent plane. Details of the physical parameter settings of the simulator have now been added in Supp (Table A).
> To evaluate the robustness of the trained policy to different masses and scales of the objects, we apply the trained policy to objects with varying masses and scales. Specifically, we vary the mass between 0.5 kg and 1.5 kg and the scale between 0.8x and 1.2x of the training size. Results can be seen in Fig. C in Supp. We observe that DexVIP remains fairly robust to large variations in these physical properties.
>
> ### 4. Functionality and posture metrics
> The functionality and posture metrics are useful to see how effectively the designed rewards translate to the learned policy, and they are meant to reflect how human-like and how poised for object use the agent’s selected poses are. We appreciate the reviewer’s point though. That is why we also report success rate and stability, which are standard metrics and agnostic to the method evaluated.
>
> **Continued below...**

---

> > ### Author Response · Authors · 2021-08-28
> > **Author Response (2/2)**
> >
> > **...continued from Response (1/2) above**
> > ### 5. Effect of predicted pose on performance
> > DexVIP uses hand poses inferred from video frames for obtaining target pose priors. As the reviewer notes, there can be errors in these predictions. To examine the effects of those errors on our policies, we train a model on ground truth (GT) hand poses from ContactPose [49] captured using mocap for all ContactDB objects. Using these GT poses to train the DexVIP policy provides an upper bound for grasp performance with perfect pose. Results are reported in Supp Table B in Sec G.1. We find that the policy trained using inferred poses performs comparably to the one trained on GT poses, showing that DexVIP is fairly robust to errors in pose predictions. We further note that the hand pose clustering process that we perform (L208-212) is able to effectively filter out bad/outlier poses so that we obtain a representative hand pose for each object (Supp Fig. D).
> >
> > ### 6. Clarifications / Other issues
> > 1. The affordance map is inferred from the image I_0.
> > 2. Here we follow the implementation of [18] exactly, so for space we did not write out those details. To recap, the distance input $d_t^r$ (L140-142) is the pairwise distance between the agent's hand and the object affordance region as defined in GRAFF [18]. Here the object affordance region is a 2D binary affordance map that is inferred from a model from [18] that is trained for affordance anticipation on ContactDB objects. Like in [18], we find that this model works reasonably well for objects outside ContactDB as well. We obtain affordance points by back-projecting the affordance map to 3D points in the camera coordinate system using the depth map at t0 to obtain M points on the hand and N points on the object (L265-267). We sample M=20 points from these back-projected points. We then track these points throughout the rest of the episode. For points on the hand, we sample N=10 regular points on the surface of the palm and fingers. The number of points at every time step remains the same across all objects. This detail has been added in the supplementary Section H. In the noise experiments (L331-339), in addition to the stated noise models, we also induce tracking failure on the affordance points to relax the tracking assumption as in [18] (described in Supp Sec A of the original submission). For computing the hand-affordance contact distance, we use the Chamfer distance between these M and N points as stated in Supp Sec B Eq 1 of the original submission.
> > 3. The object labels are obtained automatically using task id meta-data from the HowTo100M [58] videos. The task id meta-data is the search query used to collect videos for that task in HowTo100M. e.g. for a task id such as “how to buy a cast iron pan”, they gather the top N video results when YouTube is queried with this task id and assign it to all the queried videos. So no manual curation of the object type is required to gather our samples. As stated above and in L190-192 of the main paper, since we don’t require an exact match of the object instance, we find this curation of object images to be effective.
> > 4. This has been addressed above in Answer I and details have been added in Supp Sec D.
> > 5. We use the mean per-joint angle error in joint angle space of transformed FrankMocap pose expressed as robot joint angles.
> > 6. The target pose $p_{c*}^r$ is in the robot’s joint space. It is obtained through retargeting the human pose $p_{c*}^h$ from the FrankMocap joint space as described above and in Supp Sec D. Please also refer to Fig. 2 in the main paper for more clarity.
> > 7. Addressed above and in Supp Section G.1 and G.2
> > 8. This was simply for saving space.  We now report all the metrics for the ablations in the main paper in Supp Table C. We observe that the full DexVIP model gains substantially in success, stability, and posture metrics, while maintaining the functionality score of the policy that is trained using only affordance.

---

### Official Review · Reviewer_8fsL · 2021-07-24

**Originality:** Very Good
**Technical Quality:** Very Good
**Clarity Of Presentation:** Excellent
**Impact:** 4

**Recommendation:**

Strong Accept: I recommend accepting the paper and will argue for my recommendation even if other reviewers hold a different opinion.

**Summary:**

This paper proposes to guide an RL algorithm for grasping tabletop household objects with a hand pose prior derived from in-the-wild images of humans using those objects. Experiments performed in simulation show that this accelerates the learning process and makes the grasps more functional and stable.

**Issues:**

- Please discuss virtual kinesthetic teaching and challenges likely to be faced while performing sim2real transfer of the policy.
- If possible, show (preliminary) results with a policy that does not rely on RGB image input, because that modality might cause issues with sim2real transfer to a real robot.

**Reviewer Expertise:**

Excellent: Expert knowledge on the topic of the paper

**Strengths And Weaknesses:**

# Strengths
- The paper is very well written and was a joy to read!
- It presents an interesting new idea - guiding grasping algorithms with YouTube video data - and evaluates it thoroughly (albeit only in simulation).
- Models the grasping activity as a series of actions rather than the end state only (which is the formulation used by most grasping papers).
- The affordance input can also be manually generated at test time to "steer" the grasp towards a desired area of the object.
- Code will be publicly released.

# Clarification
- How is demo time calculated in Fig 6? Is it the total length of all videos from which images are chosen for DexVIP?

# Weaknesses
- Lack of real world robotic experiments is an obvious weakness, but I don't think that alone should be grounds for rejection this year.
- Effort spent in selecting an appropriate demo image may be quite high. L307 mentions that a demonstration image is collected in a few seconds. But that seems unrealistic, given that an image has to be chosen where the object is upright, the grasp is complete (not approaching), good visibility for FrankMocap, etc. Automating this process, especially detecting when the grasp is complete, might not be as straightforward as claimed. Please add some discussion and visualizations on if the proposed clustering method allows some of these considerations to be relaxed. Why not get the grasp pose from a virtual application that allows the Adroit hand joints to be interactively changed by a user (sort of like virtual kinesthetic teaching)? The paper discusses that normal kinesthetic teaching disembodies the demonstrator, but inaccuracies at a similar level might also be expected from image-based 3D pose estimation.
- It has been noted in the literature that sim2real transfer with RGB image inputs is difficult. Have you considered a policy which does not use RGB images i.e. depth + affordance only?

# Post Rebuttal
I will keep my "strong accept" rating because the rebuttal addresses my concerns. Especially the verification that the lack of RGB image input does not affect performance.

**Summary Of Recommendation:**

This paper introduces and novel idea and evaluates it as thoroughly as possible in simulation. The lack of real robot experiments and discussion on alternative demonstration methods like virtual kinesthetic teaching are drawbacks, but in my opinion not enough to warrant rejection.

---

> ### Author Response · Authors · 2021-08-28
> **Author Response**
>
> Thank you for your encouraging comments and useful suggestions!
>
> ### 1. Demo time
> Demo time in Fig. 6 is computed as the time taken to curate grasp frames from the HowTo100M dataset. For a given video from a relevant object class, we create a montage with frames sampled at regular time intervals and mark viable grasp frames. This selection process takes around 15 s per video on average, totalling 3 hrs for the curation of the entire video frame dataset, while DAPG [2] takes 30 hrs in total (Fig. 6, main paper). If we use an automatic curation process as stated in L192-194 and described below under “manual effort”, this process may be further sped up in the future.
> ### 2. Real world experiments
> To simulate a more realistic testing framework, we induce noise into the perceptual (RGBD) and proprioceptive inputs as well as the robot’s actuation (L331-339). The learned policy is shown to be quite robust to such noisy scenarios (Fig. 5, main paper). More details have been provided in Supp Sec A of the original submission. While we agree that real world evaluation would be the ultimate test for robustness, we are encouraged by the results obtained using noisy observations and actuation.
>
> ### 3. Manual effort for data curation
> We perform manual curation to extract grasp video frames in this work, and our results show that it is effective and relatively lightweight for our tests with 27 diverse objects (see above).  The hand pose clustering process that we perform (L208-212) is able to effectively filter out bad/outlier poses so that we obtain a representative hand pose for each object (Supp Fig. D).  We expect future work could accelerate curation by utilizing automatic grasp detection from methods such as 100DoH [48]. Having been trained on a large repository of YouTube videos, these models can predict pre-grasp, in-grasp, and post-grasp labels for each frame in a video. “In-grasp” labels can be used to obtain grasp images directly. Further, [48] also provides a hand pose filtering model to filter out bad hand poses. Using such automatic video processing systems could further reduce manual curation time.
>
> ### 4. Virtual kinesthetic teaching
> One of our baselines represents this approach. We compare against a model [2] that utilizes demonstrations collected from a mocap glove + VR setup to train a policy. As explained in the Introduction (L36-48) and Section 4 (L303-325), such a virtual demonstration setup with complex hardware has issues of time and cost effectiveness (Fig. 6, main paper) in addition to being a less natural mode of demo capture. In contrast, the proposed video based pose prior model effectively leverages human interactions in natural environments.
>
> ### 5. Sim2real transfer with RGB images
> Recent work shows that with domain randomization, policies trained on RGB images in simulation can work well in the real world [6, 7, 61]. Although we do not use domain randomization, following [11], we apply pixel perturbations in the range [-5, 5] on the input RGB images to simulate camera sensor noise in the real world. We find that the learnt policy remains robust to this perception noise. These are encouraging indicators for training policies on RGB images.
>
> \**update\** Per the reviewer’s request, we perform an experiment without using RGB inputs. We observe that the grasping performance for DexVIP remains the same as earlier as shown in the table below:
>
> | Model | Inputs | Success | Stability |
> |----------|----------|---------|-----------|
> |  All inputs  |   RGB + Depth + Hand-Obj Distance + Proprioception       |  68  |  50  |
> |  No RGB  |   Depth + Hand-Obj Distance + Proprioception       |  69  |  51  |
> |  No RGB-D  |   Hand-Obj Distance + Proprioception       |  41  |  25  |
> ||
>
> This indicates that although the visual stream is important, DexVIP is able to learn good grasping policies from depth even without RGB inputs, and sim2real transfer would be feasible.

---

### Meta-Review · Area_Chair_GLWX · 2021-08-13

**Recommendation:** Accept (Poster)
**Confidence:** 5

**Metareview:**

Reviewer Vrhm provides a good summary of the paper, "This work proposes an approach to learn dexterous robotic grasping from human-object interactions recorded in the video. Target hand poses are predicted from the video leveraging an existing method. The distances between target hand poses and current hand poses are used as a reward signal to train a deep reinforcement learning algorithm in the simulation. Besides the target hand poses, the affordance of each object is pre-calculated. The distances between the hand and affordance regions are also used as a reward signal. The proposed pipeline enables a 30-Dof hand to grasp 27 objects in the simulation achieving 68% success rates".

Two reviewers have recommended the paper be accepted and one has recommended a rejection. Overall, reviewers like the idea of using youtube videos to improve grasping, which I agree is an interesting area of research.  But there are several issues in the proposed method. In particular, reviewer HJKw has well summarized them:

- Run an ablation removing the object affordance reward R_aff
- Evaluate all methods only on the 11 objects not in ContactDB
- Evaluate all methods only on object classes not in ContactDB
- Clarify how object affordance maps are obtained for non-contact DB objects
- Clarify the difference between the state space between the human and robot hands

In my opinion, addressing these issues in the rebuttal is necessary. In addition, Vrhm raises concerns about "However, some important explanations of the proposed method are missing. It's unclear how the errors in the human pose prior affect the grasping performances. I would be happy to increase the evaluation score if these concerns are addressed.". Please address these concerns.

Next, it would be good to point out on what object types is the difference between GRAFF and the proposed method the maximum. I agree with reviewers Vrhm and HJKw that the grasp success rate and not posture is the appropriate metric.

Furthermore, reviewers have raised concerns about sim2real transfer. It would be worthwhile to conduct experiments with noisy observation and actuation to instill confidence that the method has a chance to work in the real world. Also, the details of physical parameters used in simulation requested by a reviewer are also critical.

**Post Rebuttal Update**
The concerns of the reviewers have been addressed. All reviewers vote to accept the paper which I agree with.

---

> ### Author Response · Authors · 2021-08-28
> **General response to meta reviewer**
>
> We thank all the reviewers for their comments and valuable suggestions. We think a misunderstanding about how the thermal images from ContactDB are used and how affordance regions are predicted may have affected the HJKw’s overall opinion on our paper. We address each of the concerns below and we accordingly updated the supplementary to include supporting details. First we briefly address the AC’s comments and point to relevant responses in the individual rebuttals below for greater clarity.
> - Run an ablation removing the object affordance reward R_aff
>     - Although our full model achieves the best performance (Fig. 5, main paper), our model improves over the GRAFF results even without using the object affordance regions, and the improvement is even larger for non-ContactDB objects. Please refer to response HJKw-2 for details.
> - Evaluate all methods only on the 11 objects not in ContactDB
>     - Grasp success and stability gains are even more substantial for non-ContactDB objects compared to other methods (ref. HJKw-3)
> - Evaluate all methods only on object classes not in ContactDB
>     - All 11 non-ContactDB objects belong to object classes not found in ContactDB, so the above result includes this case.
> - Clarify how object affordance maps are obtained for non-contact DB objects
>     - Affordance maps are obtained using the affordance prediction model from GRAFF [18] which works even for non-ContactDB objects with no thermal maps. Please refer to HJKw-1
> - Clarify the difference between the state space between the human and robot hands
>     - Please refer to Vrhm-1 for an overview. Full details have been added to Supp Sec D and Fig. B.
> - Vrhm asks how the errors in the human pose prior affect the grasping performances
>     - Our method is fairly robust to errors in pose predictions as described in Vrhm-5 and seen in Supp Table B. Moreover the clustering method filters out outlier poses as depicted in Supp Fig. D.
> - Point out on what object types is the difference between GRAFF and the proposed method the maximum. I agree with reviewers Vrhm and HJKw that the grasp success rate and not posture is the appropriate metric
>     - While we show a 13% improvement in success rate over GRAFF on ContactDB objects (Fig. 5, main paper), the gains are even more substantial at 23% for non-ContactDB objects (Fig. F, Supp). Please refer to HJKw-3 & HJKw-5 for details. Regarding metrics, we report success rate and stability, which are standard metrics and agnostic to the method evaluated. Please refer to Vrhm-4 for details.
> - Reviewers have raised concerns about sim2real transfer. It would be worthwhile to conduct experiments with noisy observation and actuation to instill confidence that the method has a chance to work in the real world.
>     - The original submission already has experiments conducted with noisy observations and actuation in Experiments Section L331-339 and Fig. 5 of the main paper with more details in Supp Sec A. We find that our method performs well in such a  noise-induced simulated environment.
> - Also, the details of physical parameters used in simulation requested by a reviewer are also critical.
>     - Please refer to Vrhm-3. This detail has been added to Supp Sec E and Table A.

---

> > ### Comment · Reviewer_8fsL · 2021-09-03
> > **discussion on the dependence on affordance output**
> >
> > HJKw has raised the issue of performance on non-contactdb objects in light of the dependance on the affordance model output. While I think the authors have addressed this in the rebuttal, I want to zoom out and present my big-picture perspective.
> >
> > The affordance model, being trained on contactdb objects and data, will perform worse on unseen objects. GRAFF, one of the baselines, depends solely on this affordance model for guidance on where to grasp.
> >
> > DexVIP (this paper) introduces a complementary source of "where to grasp" information - YouTube videos. So I would expect performance on non-contactdb object to be better than GRAFF. This has been shown in the rebuttal. I had assumed this conclusion while reviewing the paper originally, and so did not mention it. But thanks to HJKw for insisting - the response to this point makes the paper stronger in my opinion.
> >
> > The other significant baseline - DAPG - was already shown to perform worse than DexVIP on non-contactdb objects in the original paper submitted by the authors.

---

### Decision · Program_Chairs · 2021-09-13

**Decision:**

Accept (Poster)

**Comment:**

Reviewer Vrhm provides a good summary of the paper, "This work proposes an approach to learn dexterous robotic grasping from human-object interactions recorded in the video. Target hand poses are predicted from the video leveraging an existing method. The distances between target hand poses and current hand poses are used as a reward signal to train a deep reinforcement learning algorithm in the simulation. Besides the target hand poses, the affordance of each object is pre-calculated. The distances between the hand and affordance regions are also used as a reward signal. The proposed pipeline enables a 30-Dof hand to grasp 27 objects in the simulation achieving 68% success rates".

Two reviewers have recommended the paper be accepted and one has recommended a rejection. Overall, reviewers like the idea of using youtube videos to improve grasping, which I agree is an interesting area of research.  But there are several issues in the proposed method. In particular, reviewer HJKw has well summarized them:

- Run an ablation removing the object affordance reward R_aff
- Evaluate all methods only on the 11 objects not in ContactDB
- Evaluate all methods only on object classes not in ContactDB
- Clarify how object affordance maps are obtained for non-contact DB objects
- Clarify the difference between the state space between the human and robot hands

In my opinion, addressing these issues in the rebuttal is necessary. In addition, Vrhm raises concerns about "However, some important explanations of the proposed method are missing. It's unclear how the errors in the human pose prior affect the grasping performances. I would be happy to increase the evaluation score if these concerns are addressed.". Please address these concerns.

Next, it would be good to point out on what object types is the difference between GRAFF and the proposed method the maximum. I agree with reviewers Vrhm and HJKw that the grasp success rate and not posture is the appropriate metric.

Furthermore, reviewers have raised concerns about sim2real transfer. It would be worthwhile to conduct experiments with noisy observation and actuation to instill confidence that the method has a chance to work in the real world. Also, the details of physical parameters used in simulation requested by a reviewer are also critical.

**Post Rebuttal Update**
The concerns of the reviewers have been addressed. All reviewers vote to accept the paper which I agree with.